# Lysine-selective molecular tweezers are cell penetrant and concentrate in lysosomes

Zizheng Li[1], Ibrar Siddique[1], Inesa Hadrović[2], Abbna Kirupakaran[2], Jiwen Li[3], Ye Zhang [3,4,5], Frank-Gerrit Klärner[2], Thomas Schrader[2] & Gal Bitan [1,4,5✉]

Lysine-selective molecular tweezers are promising drug candidates against proteinopathies, viral infection, and bacterial biofilm. Despite demonstration of their efficacy in multiple cellular and animal models, important questions regarding their mechanism of action, including cell penetrance and intracellular distribution, have not been answered to date. The main impediment to answering these questions has been the low intrinsic fluorescence of the main compound tested to date, called CLR01. Here, we address these questions using new fluorescently labeled molecular tweezers derivatives. We show that these compounds are internalized in neurons and astrocytes, at least partially through dynamin-dependent endocytosis. In addition, we demonstrate that the molecular tweezers concentrate rapidly in acidic compartments, primarily lysosomes. Accumulation of molecular tweezers in lysosomes may occur both through the endosomal-lysosomal pathway and via the autophagy-lysosome pathway. Moreover, by visualizing colocalization of molecular tweezers, lysosomes, and tau aggregates we show that lysosomes likely are the main site for the intracellular anti-amyloid activity of molecular tweezers. These findings have important implications for the mechanism of action of molecular tweezers in vivo, explaining how administration of low doses of the compounds achieves high effective concentrations where they are needed, and supporting the development of these compounds as drugs for currently cureless proteinopathies.

[1] Department of Neurology, David Geffen School of Medicine, University of California, Los Angeles, Los Angeles, CA, USA. [2] Institute of Chemistry, University of Duisburg-Essen, Essen, Germany. [3] Department of Psychiatry and Biobehavioral Sciences, David Geffen School of Medicine, University of California, Los Angeles, Los Angeles, CA, USA. [4] Brain Research Institute, University of California, Los Angeles, Los Angeles, CA, USA. [5] Molecular Biology Institute, University of California, Los Angeles, Los Angeles, CA, USA. ✉email: gbitan@mednet.ucla.edu

Lys-selective molecular tweezers (MTs) are supramolecular ligands, first reported in 2005[1], that bind to Lys residues with low micromolar affinity and with 5–10 times lower affinity to Arg residues[2–4]. They have a horseshoe-like structure comprising hydrocarbon arms that form hydrophobic interactions with the aliphatic hydrocarbon chains of these amino-acid residues and negatively charged headgroups forming electrostatic interactions with the positive ammonium or guanidinium groups of Lys or Arg, respectively (Fig. 1a)[2–4]. MTs possess several important biological activities. The first one, originally described by Sinha et al. in 2011[5], is modulation and inhibition of the abnormal self-assembly of amyloidogenic proteins. To our knowledge, MTs are the first example of nonpeptidic, small

molecules with this type of activity, which act by a known mechanism and have well-defined binding sites. The second activity, first reported in 2015[6], is potent antiviral action against membrane-encapsulated viruses, including human immunodeficiency virus[6], Ebola[7], Zika virus[7], herpesviruses, measles virus, influenza, and SARS-CoV-2 through interaction with specific membrane proteolipid headgroups[8]. Most recently, MTs have been shown to disrupt the biofilm of the gram-positive bacterium *Staphylococcus aureus*[9].

A lead MT called CLR01 (Fig. 1b) has two phosphate groups at the central hydroquinone bridgehead. Similar to other MTs, CLR01 binds selectively to Lys residues with low micromolar affinity. It was selected as a lead compound for further

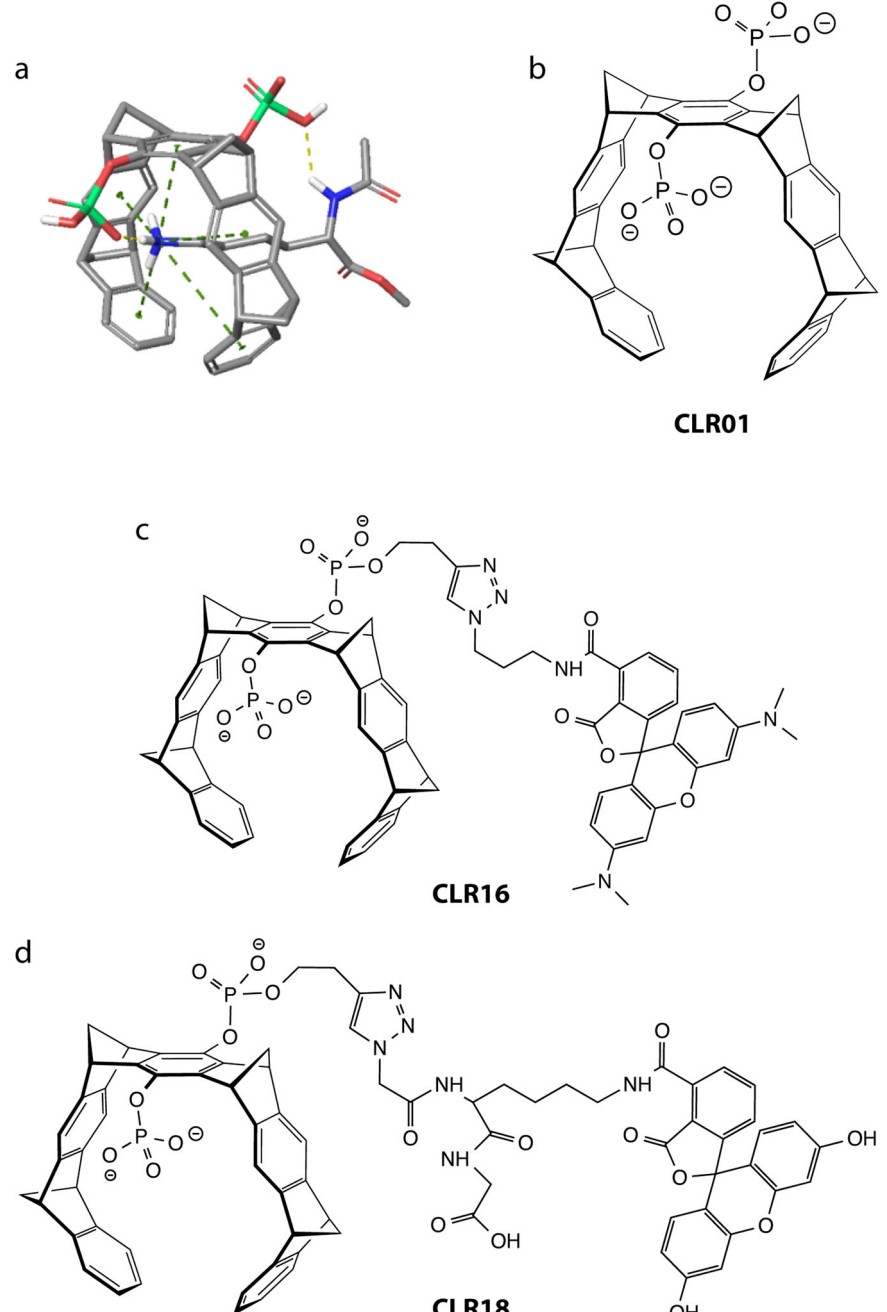

**Fig. 1 Schematic structures of MTs. a** Schematic representation of the interaction between MTs and Lys. The MT bears two headgroups that may be identical or different. The charge of each headgroup may be 0, −1, or −2, which are neutralized by the corresponding number of Na⁺ cations. **b** CLR01. **c** CLR16. **d** CLR18. All the compounds are partially protonated at pH 7.4.

development based on its low toxicity[5,10] and high activity against both amyloidogenic proteins and membrane-enveloped viruses. A key feature contributing to the low toxicity of CLR01 is its highly labile binding to Lys (and Arg) residues[11] and the fact that these residues tend to be more exposed in unstructured and misfolded polypeptides than in natively folded proteins[3]. These characteristics allow CLR01 to disrupt effectively the relatively weak hydrophobic and electrostatic interactions mediating abnormal protein self-assembly, without affecting normal physiological processes.

CLR01 has been found to inhibit the toxicity of various amyloidogenic proteins, including amyloid β-protein (Aβ), α-synuclein, islet amyloid polypeptide, and transthyretin in different cell lines and primary cultures[5,12–15]. Using rat primary hippocampal neurons, we showed that CLR01 inhibited an Aβ42-induced decrease in dendritic spine density, basal synaptic activity, and long-term potentiation[16]. Recently, we also found that CLR01 inhibited dose-dependently the prion-like propagation of tau aggregates (tau seeding) in biosensor cells[17]. Moreover, the potential therapeutic effect of CLR01 has been demonstrated in multiple animal models of various proteinopathies including the triple-transgenic mouse model of Alzheimer's disease (AD)[16], a rat model of AD[18], a mouse model of tauopathy[19], zebrafish[12,20] and mouse[15,21] models of Parkinson's disease (PD), a lamprey model of spinal cord injury[22], and mouse models of transthyretin amyloidosis[14], desmin-related cardiomyopathy[23], amyotrophic lateral sclerosis[24], multiple system atrophy[25], and the lysosomal-storage disease Sanfilippo syndrome type A[26].

Despite the promising demonstration of antiamyloid and antiviral activities in multiple cellular and animal models, until very recently, cell penetration/internalization of MTs has not been demonstrated directly. Moreover, the mechanism by which MTs may get internalized and their cellular distribution upon internalization, which may have important mechanistic implications, currently are unknown. Early findings indicated that CLR01 treatment reduced substantially the concentration of α-synuclein in a zebrafish model and suggested that this reduction was mediated through alleviation of inhibition of the ubiquitin-proteasome system (UPS) by α-synuclein oligomers[12]. A similar facilitation of UPS action was observed in a mouse model of desmin-related cardiomyopathy[23]. More recently, CLR01 was found to ameliorate significantly the pathologic phenotype in a mouse model of Sanfilippo syndrome type A by a similar relieving of "clogged" lysosomes, allowing them to resume merging with autophagosomes to form autolysosomes and degrade multiple aggregated proteins, including α-synuclein, Aβ, tau, and prion protein[26]. Based on these observations, it would be reasonable to hypothesize that MTs promote clearance of amyloidogenic proteins accumulating within cells, but how they facilitate such clearance and what cellular mechanisms might be involved is unknown.

The main impediment to answering these questions is the low intrinsic fluorescence of unlabeled MTs, which is too low to be observed on the background of a cell making studying their interaction with cells difficult. To address this challenge and provide insight into the intracellular distribution and mechanism of action of MTs, here we used two new fluorescently labeled derivatives—CLR16, labeled by 5-carboxytetramethylrhodamine (TAMRA, Fig. 1c) and CLR18, labeled by 6-fluorescein amidite (FAM, Fig. 1d). The pendant carboxylic group in the structure of CLR18 is neither required for the fluorescence label nor for the MT function. It is just a consequence of our strategy to obtain a clickable FAM label. Contrary to the TAMRA fluorophore, azido-labeled FAM was not commercially available at the time the compound was prepared for this project. Therefore, we used FAMlysine, incorporated it into a small peptide, and attached a terminal azide. Specifically, we started with a glycine-loaded Wang resin, coupled FAM-labeled lysine, and then azidoacetic acid. Cleavage from the resin released the free carboxylic acid at the C-terminus of this construct, which actually increases the water solubility of the tweezer conjugate. An inherent advantage of this strategy is the introduction of a peptidic spacer between tweezer and fluorophore. For CLR16 we purchased a TAMRA derivative with a terminal azide, allowing straightforward click reaction with an alkyne tweezer.

Recently, we used CLR16 to demonstrate that it could be internalized in a human oligodendroglioma cell line[25], yet whether the same was applicable to other cell types and the subcellular distribution of MTs remained unknown. Because the brain is particularly sensitive to proteinopathies, we used here the fluorescent MTs to explore their internalization, intracellular localization, and mechanism of endocytosis in brain cells including neurons and astrocytes.

## Results

**Characterization of CLR16 and CLR18**. The concentration dependence of the fluorescence of the two compounds was assessed by a simple titration, which showed a linear increase of the fluorescence with concentration, as expected, both for the fluorescent appendage, TAMRA at 535 nm or FAM at 518 nm (Supplementary Fig. 1a, b, respectively), and for the horseshow-like structure of the MT (335 nm). The concentration dependence of the TAMRA moiety in CLR16 (Supplementary Fig. 1a) was shallower than that of the FAM group in CLR18, allowing for measurement in the range 0.39–12.5 μM. Due the steeper concentration dependence of CLR18 (Supplementary Fig. 1b), at 0.39 μM it was similar to the blank, whereas at 12.5 μM the maximum fluorescence exceeded the maximum intensity of the fluorometer. These results suggested that both compounds could be used at 5 μM for subsequent cell-culture experiments.

To determine the pH dependence of the fluorescence, the spectrum of each compound was recorded at pH values between 3 and 8 (Supplementary Fig. 1c, d). These measurements showed distinct behaviors of the two derivatives. The fluorescence of the TAMRA group in CLR16 was maximal at pH 7 and declined when the pH was decreased or increased without an apparent change in $\lambda_{max}$ at 535 nm (Supplementary Fig. 1c). The weak fluorescence of the tweezer moiety at 335 nm decreased slightly with the decrease in pH from 8 to 3. The fluorescence of the FAM group in CLR18 at 518 nm was maximal at pH 8 and declined gradually when the pH decreased down to 5. At pH 4 or 3, the fluorescence intensity remained similar to pH 5, but the peak shifted from 518 to 570 nm (Supplementary Fig. 1d). This behavior suggested that in cell-culture experiments using a fluorescence microscope, at pH below 5, the fluorescence might not be detected because the microscope's filter only allows a window between 500 and 550 nm to be observed.

A possible concern is that the fluorescence labels might alter the tweezers' binding to Lys side chains. To test whether CLR16 and CLR18 bind Lys residue in a similar manner to CLR01, i.e., by inclusion of the Lys side chain within the tweezer's cavity, we performed NMR (Supplementary Fig. 2) and fluorescence (Supplementary Figs. 3 and 4) titrations with Ac-Lys-OMe. The NMR experiments, which were done at high concentration, showed upfield shifts of the Lys side chain methylene groups indicating inclusion inside the tweezer's cavity, yet line broadening suggested that at these concentrations the labeled tweezers form clusters. In contrast, the fluorescence titrations used the tweezers at low μM concentrations, similar to the cell-culture experiments (see below), and showed significant quenching of the inherent tweezer fluorescence indicating that CLR16 and CLR18

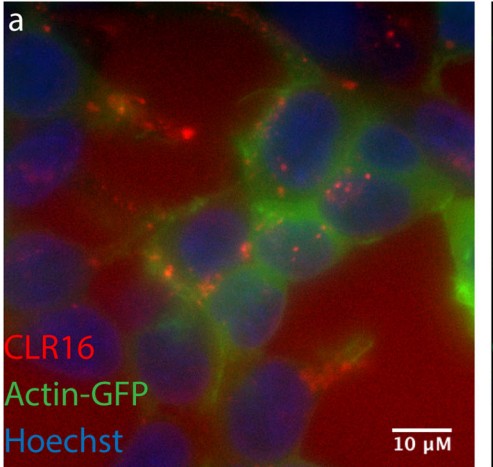
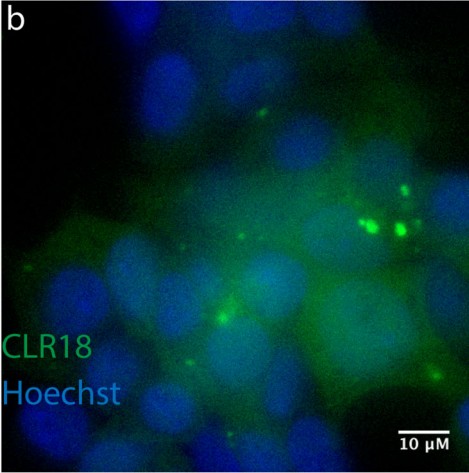

**Fig. 2 MTs are internalized in SH-SY5Y cells and concentrate in puncta.** SH-SY5Y cells were incubated with fluorescent MTs for 24 h and visualized by fluorescence microscopy. **a** Cells were transiently transfected with GFP-actin (green), incubated with 5 μM CLR16 (red), and nuclei were stained with Hoechst (blue). **b** Cells were incubated with 10 μM CLR18 (green) and nuclei were stained with Hoechst (blue).

bound to the Lys side chain by a similar binding mode to CLR01 (Fig. 1a). Interestingly, however, the binding of the fluorescent tweezers had higher affinity, in the nanomolar range, than that of CLR01, and a second, weak binding of Ac-Lys-OMe was observed, likely to the fluorescent tag itself.

**Fluorescently labeled MTs are internalized by neurons and astrocytes.** To test whether MTs are internalized by brain cells other than oligodendrocytes, first, we incubated undifferentiated human neuroblastoma SH-SY5Y cells with CLR16 or CLR18 for 24 h and examined the cells by fluorescence microscopy. Although the fluorescence spectra of CLR16 and CLR18 showed that their $\lambda_{max}$ were relatively close (Supplementary Fig. 1), for the cell-culture experiments, we used the microscope filters according to the reported $\lambda_{max}$ of absorption and emission of the TAMRA ($\lambda_{ab} = 555$, $\lambda_{em} = 580$) and FAM ($\lambda_{ab} = 494$, $\lambda_{em} = 518$) groups. To visualize the cell body and nucleus, in experiments using the red-fluorescent CLR16 we expressed actin conjugated to green fluorescent protein (actin-GFP) and stained the cells with Hoechst dye, respectively. In the case of CLR18, the green fluorescence of the fluorescein dye precluded the use of actin-GFP and only nuclei were stained. In both cases, we observed internalization of the dye in the cells (Fig. 2 and Supplementary Movie 1). Interestingly, outside the cells, each dye yielded diffuse fluorescence in the media, whereas inside the cells, in addition to similar diffuse fluorescence in the cytoplasm, CLR16 (Fig. 2a) and CLR18 (Fig. 2b) appeared as bright red or green puncta, respectively. Neither compound was found in the nucleus. The punctate appearance of CLR16 and CLR18 in the treated cells suggested either that the dyes formed large aggregates or that they concentrated in particular cellular compartments/organelles. Previously, CLR01 was found to form weak dimers at high concentrations but not large aggregates or colloids[6,27]. Therefore, assuming a similar behavior by CLR16 and CLR18 we hypothesized that they concentrated in certain organelles and asked what these organelles might be.

To explore potential answers, we treated the cells with CLR16, as described above, and costained them with markers of mitochondria, early endosomes, late endosomes, autophagosomes, or lysosomes. The colocalization of MT puncta and organelles was quantified using the method of Manders et al.[28] (Supplementary Fig. 5). Most of the experiments were done only with CLR16, for convenience of selecting the appropriate

wavelength filters, whereas CLR18 was used only in a few cases for validation.

First, we incubated the cells with CLR16 in the presence of actin-GFP and Hoechst stain, and added MitoTracker™ Deep Red FM to visualize mitochondria. These experiments showed little to no colocalization of CLR16 with mitochondria (Fig. 3a–c, Supplementary Fig. 5a). Similar experiments using the early- and late-endosome markers Rab5a and Rab7a, respectively, showed minimal colocalization of CLR16 with early endosomes (Fig. 3d–f, Supplementary Fig. 5a) and increased colocalization with late endosomes (Fig. 3g–i, Supplementary Fig. 5a). The highest colocalization was found with lysosomes, for which the overlap between CLR16 and the dye LysoTracker™ was nearly complete (Fig. 4a–c, Supplementary Movie 2, Supplementary Fig. 5a). A similar analysis using CLR18 showed that nearly every punctum of the compound in SH-SY5Y cells overlapped with a lysosome. However, unlike CLR16, CLR18 fluorescence overlapped only with a fraction of the lysosomes (Fig. 4d–f, Supplementary Fig. 5b), likely reflecting the shift in $\lambda_{max}$ from 518 to 570 nm (Supplementary Fig. 1d), reducing substantially the detection of the compound using the green filter of the microscope.

SH-SY5Y is a convenient cell line for initial analysis, but the cells are only an approximation of neurons, especially in their nondifferentiated form. Therefore, we examined next the internalization and colocalization of CLR16 with lysosomes in mouse primary hippocampal neurons. Because primary neurons are more fragile than neuroblastoma cells, we did not label them with actin-GFP and rather used brightfield images to observe the cell body (Fig. 4g–i, Supplementary Fig. 5c). The images showed that CLR16 colocalized with lysosomes in the soma of the neurons, similar to its behavior in SH-SY5Y cells (Fig. 4a–c).

To test whether MTs behave similarly with other brain cell types, we incubated mouse primary astrocytes with LysoTracker™ for 16 h, then added CLR16, incubated for an additional 12 h, and visualized the cells. These experiments showed colocalization of CLR16 with lysosomes (Supplementary Figs. 5c and 6), similar to the data observed in neurons.

**Cellular uptake of CLR16.** We asked next how MTs are taken up by cells and what path they undergo before ending up in the lysosomes. Because the negatively charged groups of the MTs may interfere with passive permeation through the cell

### Mitochondria

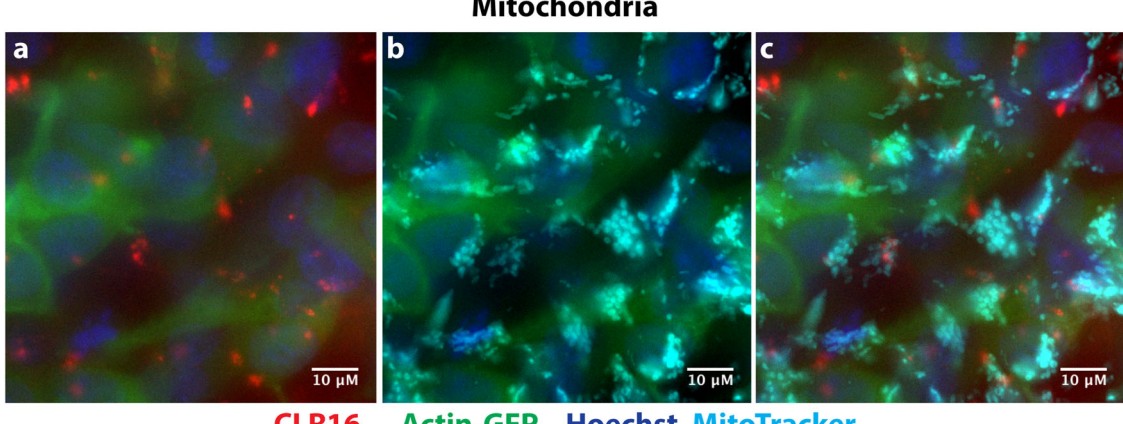

**CLR16** **Actin-GFP** **Hoechst** **MitoTracker**

### Early Endosomes

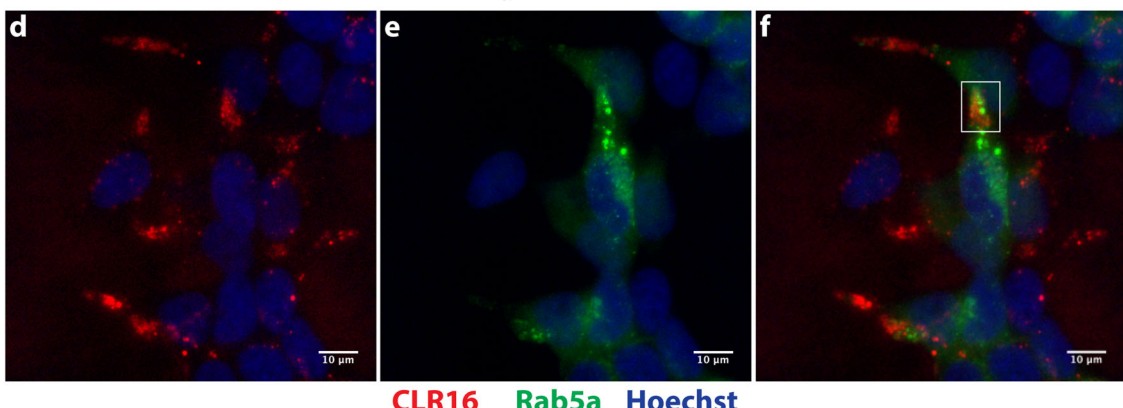

**CLR16** **Rab5a** **Hoechst**

### Late Endosomes

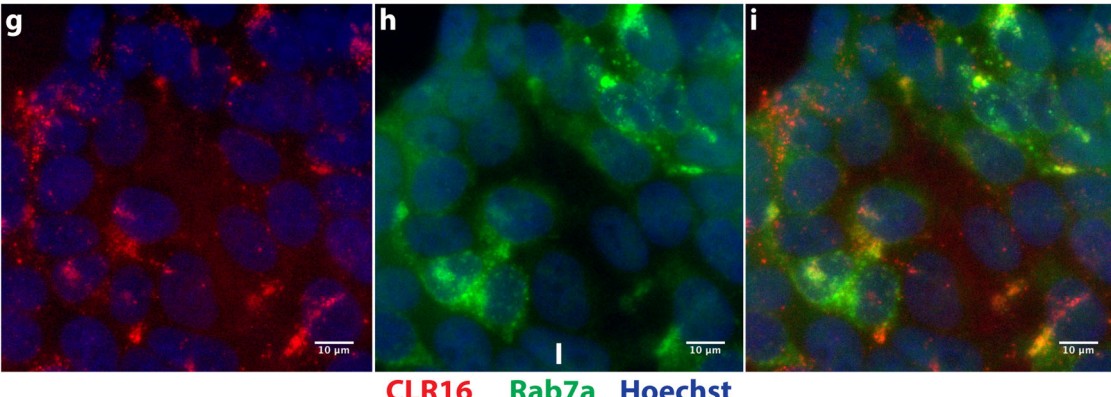

**CLR16** **Rab7a** **Hoechst**

**Fig. 3 CLR16 colocalizes partially with endosomes but not mitochondria.** SH-SY5Y cells were incubated with CLR16 for 24 h in the presence of specific markers for the different organelles. **a** Cells were transiently transfected with GFP-actin, incubated with 5 μM CLR16, and nuclei were stained with Hoechst. **b** Mitochondria were stained with MitoTracker™ Deep Red FM. **c** Overlap of **a** and **b** shows that CLR16 does not colocalize with mitochondria. **d** Cells were incubated with 5 μM CLR16 and nuclei were stained with Hoechst. **e** Early endosomes were visualized by transient transfection with Rab5a-GFP. **f** Overlap of **d** and **e** shows that CLR16 colocalizes minimally with early endosomes (highlighted in a white box). **g** Cells were incubated with 5 μM CLR16 and nuclei were stained with Hoechst. **h** Late endosomes were visualized by transient transfection with Rab7a-GFP. **i** Overlap of **g** and **h** shows that CLR16 colocalizes partially with late endosomes.

membrane, we hypothesized that MTs are internalized via active endocytosis. To investigate this possibility, we incubated SH-SY5Y cells with CLR16 in the absence or presence of increasing concentrations of the dynamin inhibitor dynasore, which specifically blocks dynamin-dependent endocytosis. Dynasore is also known to inhibit micropinocytosis[29,30] by disrupting lipid rafts[31] and membrane ruffling[30], as well as destabilizing F-actin[30]. If CLR16 internalization requires active dynamin-dependent endocytosis, it would be expected to decrease dose-dependently with increasing dynasore concentration.

**CLR16 - SH-SY5Y cells**

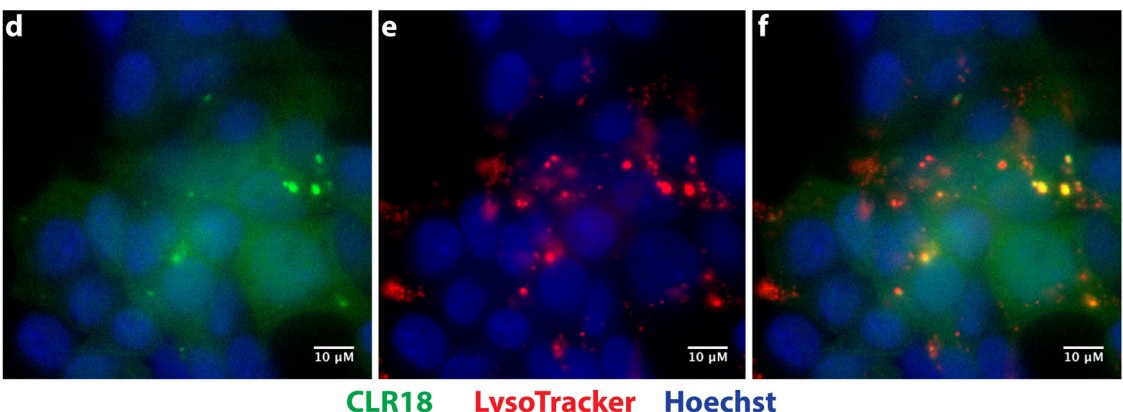

CLR16    Actin-GFP    Hoechst    LysoTracker

**CLR18 - SH-SY5Y cells**

CLR18    LysoTracker    Hoechst

**CLR16 - primary mouse hippocampal neurons**

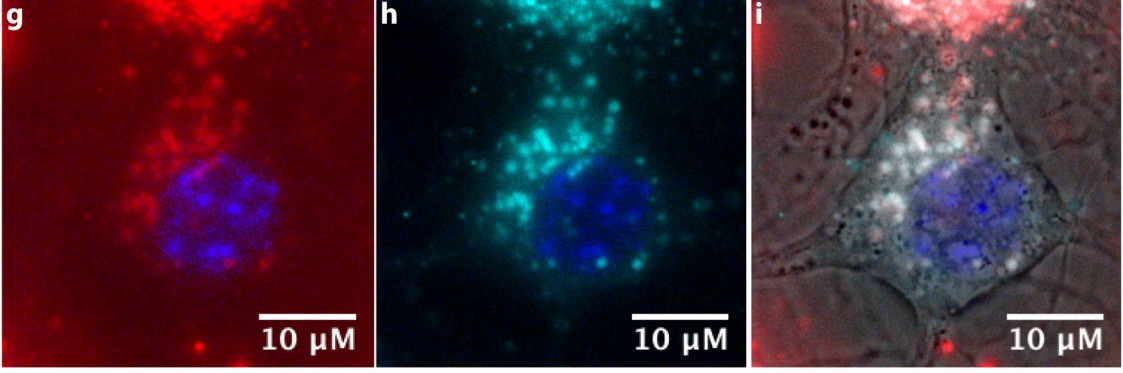

CLR16    Hoechst    LysoTracker

**Fig. 4 MTs colocalize strongly with lysosomes in neurons. a** SH-SY5Y cells were transiently transfected with GFP-actin, incubated with 5 μM CLR16 for 24 h, and nuclei were stained with Hoechst. **b** Lysosomes were stained with LysoTracker™ (pseudo-colored cyan). **c** Overlap of **a** and **b** shows strong colocalization of CLR16 with lysosomes. **d** SH-SY5Y cells were incubated with 10 μM CLR18 for 24 h and nuclei were stained with Hoechst. **e** Lysosomes were stained with LysoTracker™ (pseudo-colored red). **f** Overlap of **d** and **e** shows colocalization of CLR18 with lysosomes. **g** Primary mouse hippocampal neurons were incubated with 10 μM CLR16 for 24 h and nuclei were stained with Hoechst. **h** Lysosomes were stained with LysoTracker™ (pseudo-colored cyan). **i** Overlap of **g** and **h** shows the colocalization of CLR16 with lysosomes. The neurons' morphology is shown by overlap with the brightfield image.

To assess such an effect, we expressed the late-endosome marker Rab7a, as described above, and imaged the cells at different time points after adding CLR16. We verified that under the experimental conditions, reduction of fluorescence did not reflect cytotoxicity by dynasore (Supplementary Fig. 7). In addition to labeling late endosomes, the diffuse fluorescence of Rab7a in the cytoplasm (Fig. 3h) helped delineate the cell boundaries.

In the absence of dynasore, at $t = 5$ h, both diffuse and punctate CLR16 fluorescence were observed (Fig. 5a), which

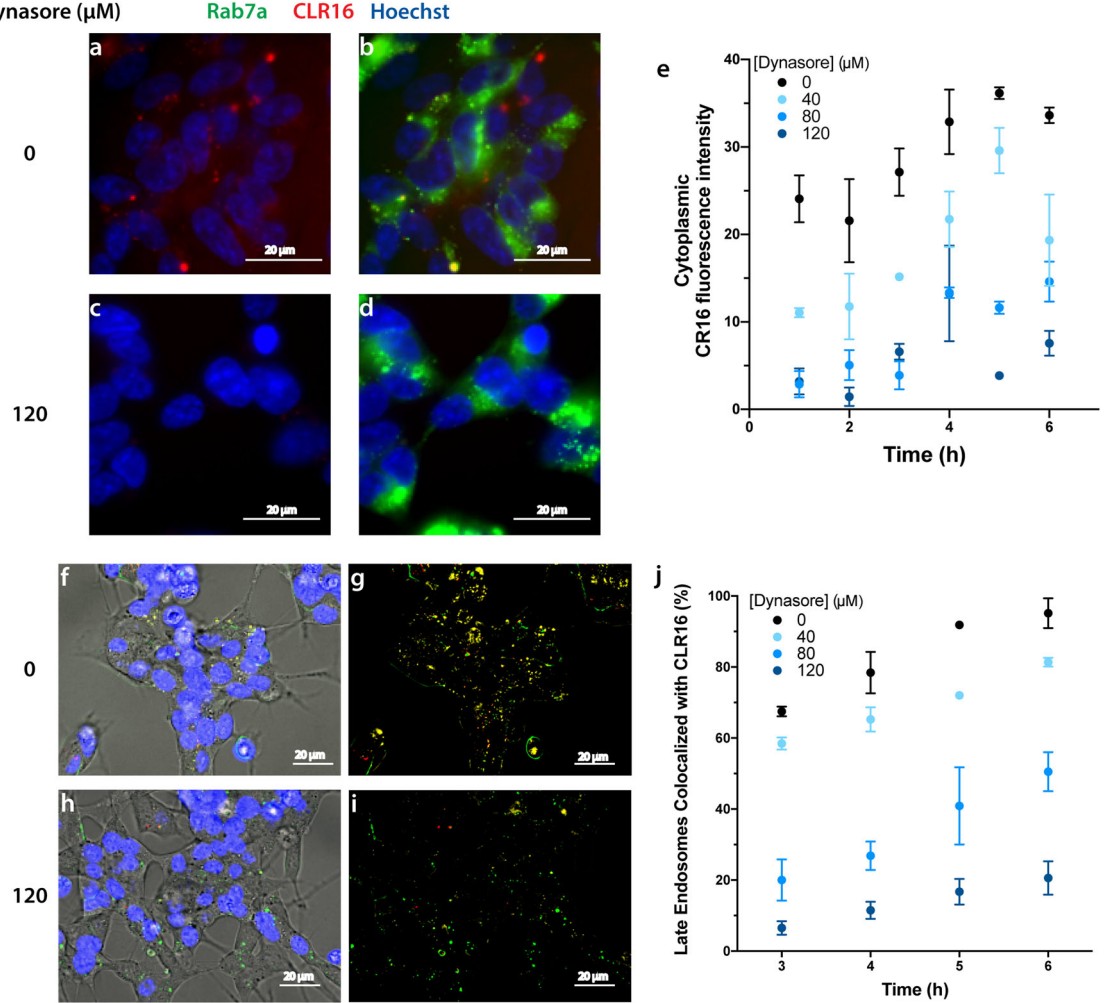

**Fig. 5 Dynasore inhibits CLR16 internalization.** SH-SY5Y cells were transfected with Rab7a-GFP, and incubated with 5 μM CLR16 and different concentrations of dynasore. **a**–**d** Representative images at $t = 5$ h after adding CLR16. **e** Quantification of the average fluorescence intensity of CLR16 ($N = 3$ arbitrarily chosen fields of view in the same experiment). Fluorescent images obtained using haze-reduction settings either overlapping with brightfield images (**f**, **h**) or alone (**g**, **i**) in the absence (**f**, **g**) or presence (**h**, **i**) of 120 μM dynasore. **j** Quantification of the percentage of late endosomes overlapping with CLR16 in all the late endosomes ($N = 3$ arbitrarily chosen fields of view in the same experiment). The data are presented as mean ± SD.

colocalized to a large extent with Rab7a-GFP fluorescence (Fig. 5b), though the diffuse cytoplasmic fluorescence of CLR16 and Rab7a-GFP partially obscured the puncta. In contrast, in the presence of 120 μM dynasore, little or no CLR16 fluorescence was observed in the cells (Fig. 5c, d). Quantitation of the average CLR16 fluorescence intensity in the cells confirmed a time-dependent increase, which was reduced dose-dependently by dynasore at each time point (Fig. 5e). To further quantify the colocalization of CLR16 and late endosomes, the diffuse fluorescence in both the green and the red channels was removed by using haze-reduction settings in the microscope software. Under these conditions, substantial overlap of CLR16 and Rab7a-GFP was observed in the puncta in the absence of dynasore (Fig. 5f, g, yellow), whereas in the presence of 120 μM dynasore the vast majority of the puncta were green (Fig. 5h, i), demonstrating the inhibition of CLR16 endocytosis. Quantitation was performed by counting the puncta manually and calculating the fraction of late endosomes colocalized with CLR16 (yellow puncta) in all the late endosomes (yellow + green puncta) in representative fields of view as a function of time and dynasore concentration. In all cases, we found a time-dependent increase in the percentage of late endosomes colocalized with CLR16 between 3 and 6 h, which was reduced dose-dependently by

dynasore (Fig. 5j). These results suggested that dynamin-dependent endocytosis is a major pathway by which CLR16 entered the cells, although partial entry by clathrin-mediated endocytosis, passive diffusion, or other mechanisms could not be ruled out.

Few puncta showed red fluorescence only, suggesting that during the time frame of this experiment, 6 h, Rab7a marks both late endosomes and endolysosomes formed by fusion of the late endosomes with the lysosomes. This is in contrast to the partial colocalizaion of CLR16 with late endosomes observed at 18 h (Fig. 3g–i). Presumably, at the later time point, Rab7a had been recycled to late endosomes and no longer colocalizes with lysosomes.

To further explore the CLR16 endocytosis pathway, we measured the time-dependence of CLR16 colocalization with early endosomes, late endosomes, and (endo)lysosomes for 18 h using live-cell imaging. To secure enough time for microscope setup, the first time point was taken 30 min after adding CLR16 and to minimize cellular damage caused by the fluorescent-microscope laser, the image-capture interval was set to 20 min. The degree of CLR16 colocalization with the cytoplasm, early endosomal, late endosomal, and lysosomal compartments was defined first using the appropriate marker and then CLR16

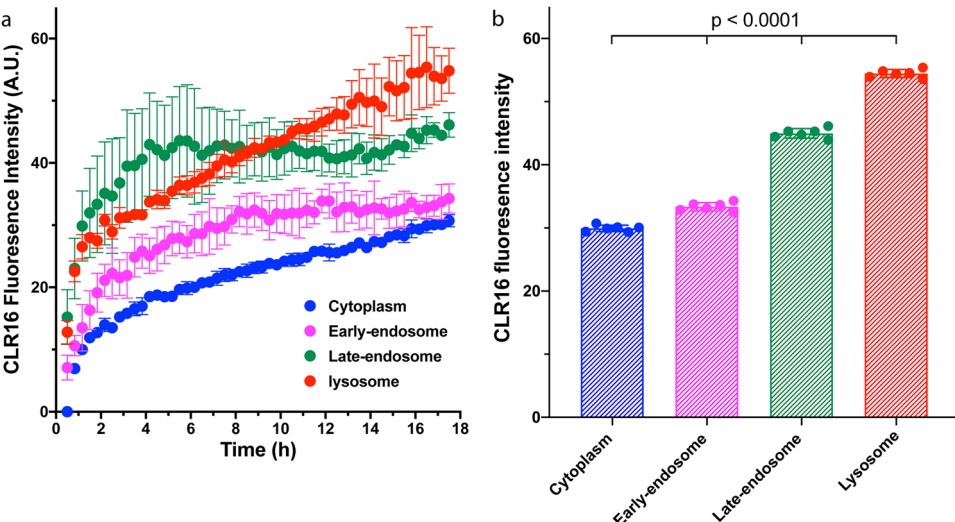

**Fig. 6 Time-lapse quantification of CLR16 fluorescence at lysosomes, early endosomes, and late endosomes.** SH-SY5Y cells were either transfected with Rab5a-GFP or Rab7a-GFP, or treated with LysoTracker™, to define the area of early endosomes, late endosomes, or lysosomes, respectively. **a** Five micromolar CLR16 was added and time-lapse images were captured every 20 min. The mean CLR16 fluorescence in each cell compartment was quantified ($N = 3$ arbitrarily chosen fields of view in the same experiment). **b** The maximal fluorescence was calculated as the average of the last 2 h of measurement ($N = 6$ time points). The data are shown as mean ± SD. $P$ values were calculated by a one-way ANOVA.

fluorescence intensity was measured within the regions of overlap (the cytoplasm was defined as the intracellular diffuse fluorescence excluding bright puncta). The first time point in the cytoplasm was set as baseline, which was subtracted from all other data points.

At the earliest time point, 30 min, the highest fluorescence was in the Rab7a-positive puncta, corresponding to late endosomes/endolysosomes, which was about twice the signal in the Rab5a-positive puncta (early endosomes) and ~25% higher than in the LysoTracker™-positive puncta (Fig. 6a), suggesting that as CLR16 gets internalized, it traffics quickly from the early endosomes to the late endosomes, and at a slower rate to the lysosomes, likely via fusion of late endosomes with lysosomes to form endolysosomes. Thus, the colocalization of CLR16 with Rab7a plateaued after ~4 h under these experimental conditions, whereas the colocalization with LysoTracker™ continued to increase and surpassed that of Rab7a at ~8 h. At this time point, the fluorescence in the early endosomes also plateaued, whereas in the LysoTracker™-positive puncta, the CLR16 fluorescence reached a plateau value only at 16 h (Fig. 6a). The maximal fluorescence was calculated as the average of the last 2 h of measurement and was 29.9 ± 0.5 in the cytoplasm, 33.4 ± 0.7 in the early endosomes, 45.0 ± 0.8 in the Rab7a-positive late endosomes/endolysosomes, and 54.4 ± 0.6 in the LysoTracker™-positive lysosomes (Fig. 6b).

**CLR16 colocalizes with autophagosomes.** As drug candidates for proteinopathies, MTs are thought to exert their therapeutic effect by binding primarily to Lys residues in misfolded proteins and inhibiting their self-assembly. One of the ways by which cells remove misfolded proteins is by sequestering them in autophagosomes, which then fuse with lysosomes to form autolysosomes where these proteins are to be degraded[32]. Therefore, we asked if in addition to accumulating in endolysosomes via the endosome-lysosome pathway, as discussed above, MTs also might be present in autophagosomes. To explore this possibility, we expressed GFP-tagged p62 as a marker of autophagosomes in SH-SY5Y cells and treated the cells with CLR16 in the absence or presence of chloroquine, which inhibits the fusion of autophagosomes with lysosomes, leading to accumulation of autophagosomes. Live-cell,

time-lapse imaging showed that in the absence of chloroquine, although CLR16 puncta were apparent (Fig. 7a, c) the time resolution of the experiment was not sufficient for observing the autophagosomes (Fig. 7b, c), whereas visualization in the presence of chloroquine showed clearly that CLR16 colocalized with autophagosomes, although to a lower extent than with lysosomes (Fig. 7d–f, Supplementary Fig. 5a).

**CLR16 colocalizes with tau aggregates in lysosomes.** In a recent study[17], we showed that CLR01 inhibited tau seeding in biosensor HEK293 cells expressing the four-repeat domain of tau carrying the disease-associated P301S substitution and conjugated to cyan fluorescent protein or yellow fluorescent protein[33]. However, because CLR01's fluorescence is too weak, as discussed above, its interaction with tau in the biosensor cells could not be demonstrated. Here, to test whether MTs colocalize with tau aggregates, we treated the same biosensor cells with CLR16 after inducing intracellular tau aggregation using a brain extract of an 8-month-old PS19 mouse. The PS19 line is a tauopathy mouse model expressing human tau carrying the same P301S substitution. Fluorescence microscopy showed that the cells contained both tau puncta (Fig. 8a) and large tau aggregates (Fig. 8e). CLR16 colocalized with the puncta (Fig. 8b, d, arrowheads), but not with the large aggregates (Fig. 8f, h, arrows). The puncta also colocalized with lysosomes (Fig. 8c, d), whereas the large aggregates did not (Fig. 8g, h), suggesting that MTs facilitate tau clearance by dissociating tau oligomers and small aggregates primarily in lysosomes, rather than by a direct effect on large, insoluble tau aggregates, such as neurofibrillary tangles.

**Indirect evidence for colocalization of CLR01 with lysosomes.** The TAMRA and FAM fluorophores render CLR16 and CLR18, respectively, larger and more hydrophobic than CLR01 (Fig. 1), which may affect their ability to enter cells and their intracellular localization compared to CLR01. Therefore, to gain insight into the putative internalization and subcellular localization of CLR01, we asked whether competition of the fluorescently labeled tweezers by the unlabeled CLR01 could provide indirect evidence for endocytosis of CLR01 and a possible preference for lysosomal localization. As the binding titrations (Supplementary Fig. 3)

Chloroquine

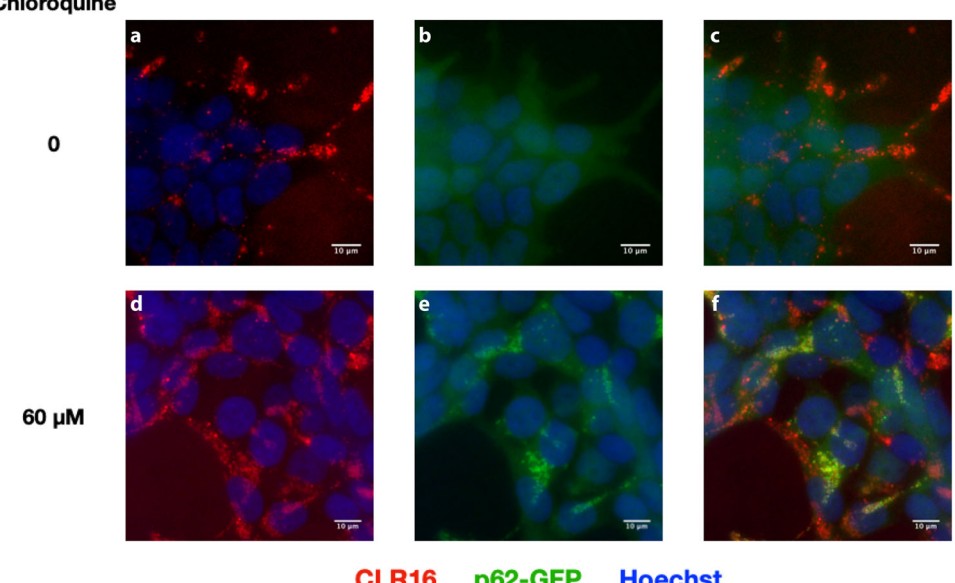

**Fig. 7 CLR16 colocalizes with autophagosomes.** SH-SY5Y cells were transfected with GFP-p62, incubated with 5 µM CLR16 in the absence (**a**–**c**) or presence (**d**–**f**) of chloroquine for 24 h, and stained with Hoechst.

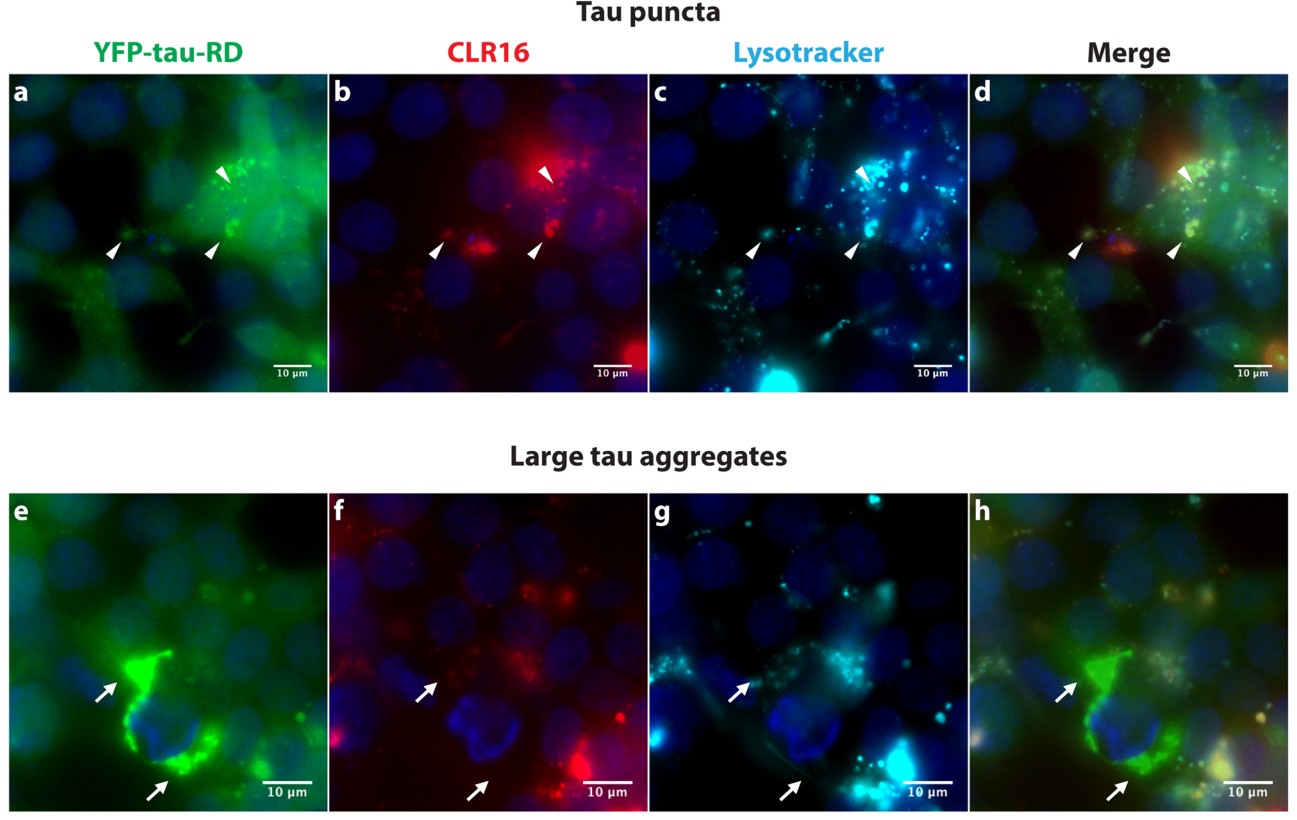

**Fig. 8 CLR16 colocalizes with tau aggregates in lysosomes.** Biosensor HEK293 cells were treated with a brain extract from a PS19 transgenic mouse to induce tau aggregation, Then, 5 µM CLR16 and LysoTracker were added. **a**–**d** CLR16 colocalized with small tau puncta at lysosomes (white arrowheads). **e**–**h** CLR16 did not colocalize with large tau aggregates (white arrows).

suggested that CLR16 bound Lys more avidly than CLR01, we expected the competition to be weak, which means that a relatively large excess of CLR01 would be needed to observe such a competition. Thus, we conducted live-cell, time-lapse imaging of SH-SY5Y cells cotreated with LysoTracker™, CLR16, and up to 40 times higher concentrations of CLR01 with monitoring of the LysoTracker™ and CLR16 fluorescence. If CLR01 competes with CLR16 for entering cells and/or lysosomes, the degree of overlap between LysoTracker™ and CLR16 would decrease with increased CLR01 concentration.

As a precaution, we tested first whether CLR01 might quench the fluorescence of CLR16 in vitro. We titrated CLR01 into a

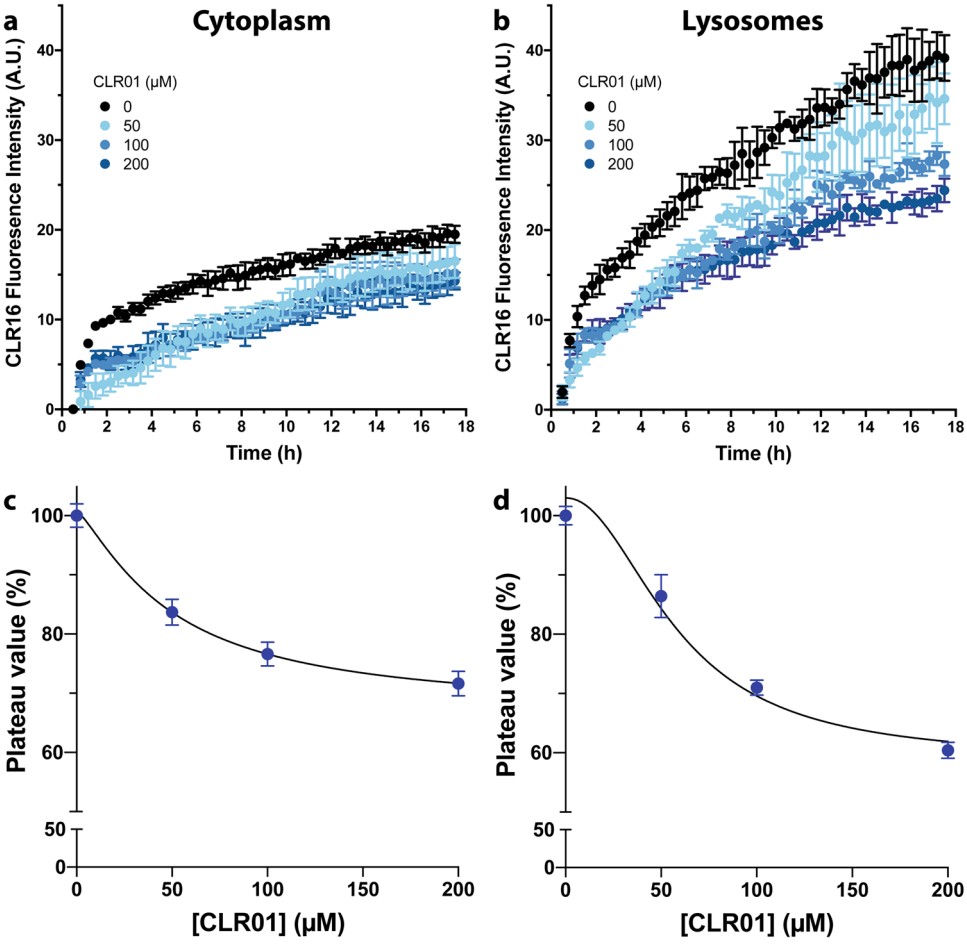

**Fig. 9 Excess CLR01 competes with cell internalization and lysosomal localization of CLR16.** SH-SY5Y cells were transfected with actin-GFP and treated with LysoTracker™ to visualize the cytoplasm and lysosomes, respectively. Cells were pretreated with different concentrations of CLR01 4 h prior to the addition of 5 μM CLR16. Time-lapse images of three fields of view for each time points were measured every 20 min and averaged. Cytoplasmic (**a**) and lysosomal (**b**) CLR16 fluorescence were quantified ($N = 3$ arbitrarily chosen fields of view in the same experiment). The plateau fluorescence of cytoplasmic (**c**) and lysosomal (**d**) CLR16 was calculated as the average of the last 2 h of measurement ($N = 6$ time points). The percentage of fluorescence intensity compared to control cells not treated with CLR01 is shown. The data are presented as mean ± SD.

5-μM solution of CLR16 in cell-culture medium and measured TAMRA emission using a fluorometer (Supplementary Fig. 8a). CLR01 quenched the fluorescence of CLR16 weakly, reaching ~10% intensity reduction at 40-fold excess (Supplementary Fig. 8b). With that in mind, to determine the effect of CLR01 on CLR16 fluorescence in the SH-SY5Y cells, we quantified the mean TAMRA fluorescence intensity in the area defined by the actin-GFP signal (cytoplasm) and separately in the area defined by the LysoTracker™ fluorescence (lysosomes) at each time point (Fig. 9a, b). The plateau value was calculated as the average fluorescence in the last 2 h of the measurement and was normalized to the control condition (no CLR01 added, Fig. 9c, d). The plateau CLR16 fluorescence was reduced by nearly 30% in cytoplasm, and 40% in the lysosomes in the presence of 40-fold excess CLR01. These results indicated that only part of the reduction in fluorescence could be attributed to the quenching observed in cell-culture medium, suggesting that CLR01 competed with CLR16 for entry into the cells and colocalization with lysosomes.

## Discussion
MTs are promising drug candidates for proteinopathies showing beneficial therapeutic effects in multiple cell-culture and animal models[2–4]. Entry of these compounds into cells has been

postulated but not demonstrated until very recently, and their intracellular distribution has been unknown. Recently, we showed that CLR16 crossed the plasma membrane and appeared as puncta in the cytoplasm of oligodendrocytes[25]. Here, we demonstrate that CLR16 and a related derivative, CLR18, enter neuronal SH-SY5Y cells, HEK293 cells, primary mouse hippocampal neurons, and primary mouse astrocytes (Figs. 2, 4, 5, and 9, Supplementary Movie 1, Supplementary Fig. 5). Moreover, we show that shortly after addition to the medium, upon entry into the cells, the compounds accumulate in acidic compartments, primarily lysosomes (Figs. 4, 5, and 9, Supplementary Movie 2, and Supplementary Fig. 5). We also found that after 18 h of incubation with SH-SY5Y cells, CLR16 colocalized weakly with early endosomes (Fig. 3d–f) and moderately with late endosomes (Fig. 3g–i). Kinetic analysis suggested that CLR16 might accumulate first in the late endosomes and gradually move to the lysosomes until its concentration in the lysosomes surpasses the concentration in the late endosomes (Fig. 6). Alternatively, the kinetic data, in combination with the dynasore-inhibition experiments (Fig. 5), suggest a transition from the late endosomes to endolysosomes, which still contain Rab7a at 6 h (Fig. 5), but not after 18 h of incubation (Fig. 3g–i). CLR16 was found to colocalize partially also with autophagosomes (Fig. 7), suggesting that it may accumulate in lysosomes via two different pathways—

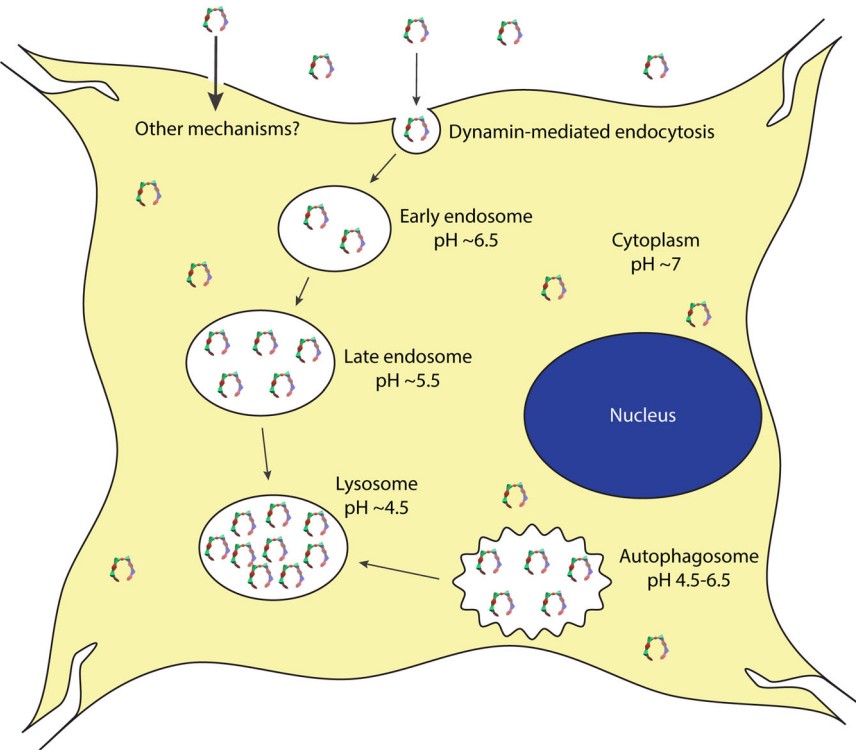

**Fig. 10 Schematic summary of MTs internalization and cellular distribution.** MTs enter the cells via dynamin-dependent endocytosis and potentially other mechanisms. They then move from early endosomes to late endosomes to lysosomes, where they reach the highest concentration. Alternatively, they also may end up in autolysosomes after initially colocalizing with autophagosomes.

endocytosis leading from the extracellular space to early endosomes, late endosomes, and endolysosomes, or after entering the cell, autophagosomes, and then autolysosomes (Fig. 10).

These pathways are characterized generally by a gradual decrease in pH. The extracellular space and cytoplasm have a neutral or slightly basic pH (7.0–7.4). The pH of early endosomes is ~6.5, that of late endosomes is ~5.5, and the lysosomes have a pH of ~4.5[34,35]. Autophagosome have a wider range of pH, from 4.5 to 6.5. Our results suggest that the accumulation of CLR16 correlates with a decrease in pH. Interestingly, the fluorescence intensity of the compound decreases with the decrease in pH, suggesting that the CLR16 fluorescence detected in these organelles is an underrepresentation of its actual concentration. Accumulation in acidic compartments is typical for weakly basic, lipophilic compounds, e.g., hydrophobic amines, which are known to diffuse through the membrane into acidic cellular compartment in their neutral state and get trapped inside due to protonation that renders them ionic and no longer membrane permeable[36]. MTs have the opposite chemical character, they are negatively charged at moderately acidic and basic pH and become neutral at pH ~ 2. Thus, their trapping at the moderately acidic pH of the lysosomes may seem paradoxical. A plausible explanation for this trapping is the binding of MTs to Lys residues in proteins or peptides that become trapped in the acidic compartments, though the exact mechanism will require further investigation.

The presence of CLR16 and CLR18 in the cytoplasm already at the earliest time points we measured, together with the data demonstrating internalization via dynamin-dependent endocytosis, suggests either that a fraction of the compounds could diffuse into the cell directly through the plasma membrane or that upon entry through endocytosis, a fraction of the endocytosed MT escapes from the organelles into the cytoplasm. Another possibility is receptor-mediated, dynamin-independent pathways

of cell entry. In support of the existence of alternative pathways, CLR16's cell entry and lysosomal accumulation was not blocked completely even at the highest concentration of dynasore.

The goal of our study was to elucidate the cellular localization of MTs, most of which, such as CLR01, cannot be observed directly because their intrinsic fluorescence is too weak. However, the large and hydrophobic TAMRA and FAM fluorophores in CLR16 and CLR18, respectively, may affect the cellular distribution differently from smaller and more hydrophilic MTs. To address this question, we tested whether CLR01 might compete with CLR16 for the same cellular compartments, as a way to observe indirectly the localization of CLR01. As CLR16 binds more avidly to Lys than CLR01, competition experiments are difficult and require a large excess of the competitor, yet we are not aware of better alternatives for addressing this question. We found that the cytoplasmic and the lysosomal CLR16 fluorescence were reduced by ~30% and ~40%, respectively, in the presence of 40-fold excess CLR01 (Fig. 9b, d). We also found that CLR01 quenches CLR16's fluorescence, but by only up to 10% (Supplementary Fig. 8). Our results show that CLR01 partially displaced CLR16 under these conditions, and therefore a plausible interpretation is that CLR01 behaves similarly to CLR16 in terms of cell entry and lysosomal accumulation.

Multiple studies have shown that misfolded protein seeds are internalized via endocytosis in proteinopathies[37–39]. Degradation of misfolded proteins by endolysosomes is important for inhibiting cell-to-cell, prion-like spreading of the seeds[38,40]. Another important mechanism for degradation of abnormal protein aggregates is autophagy[32]. However, in proteinopathies, the cellular protein degradation and clearance mechanisms gradually become overwhelmed by the accumulating misfolded and aggregated proteins and eventually fail, leading to cell degeneration and death[41]. MTs have been shown to inhibit seeding and toxicity in cell culture[3,17] and reduce the accumulation of the

offending proteins in multiple animal models[4,15,18,21,23–25]. However, it has been unclear how very low concentrations of the compounds, particularly in the brain[10], still yield substantial therapeutic effects.

In the first mouse study examining CLR01, 28-day sub-cutaneous administration of the compound in the triple-transgenic mouse model of AD[42] at 0.04 mg kg$^{-1}$ per day, which was estimated to yield low nM to high pM concentrations in the brain[10], led to a robust reduction in both amyloid plaques and neurofibrillary tangles[16]. Similarly, treatment of a mouse model of PD with the same dose of CLR01 administered sub-cutaneously twice weekly led to rescue of dopaminergic neurons and significant amelioration of motor deficits[15]. Part of the explanation is accumulation of the compound in the brain over time due to a brain-clearance rate that is much slower than the clearance of the compound from the blood[10], yet a remaining question has been how MTs achieve sufficient effective concentrations to prevent aberrant protein self-assembly and facilitate clearance of the rogue protein oligomers and aggregates in the context of a cell, where so many competing Lys/Arg binding sites are present.

Part of the answer has been provided recently by revisiting the question of the blood–brain barrier penetration of CLR01. Previously, following intravenous administration, the brain-to-blood ratio was found to be 2–3% in wild-type mice, the triple-transgenic mouse model of AD[10], and the Thy1-aSyn model of PD[21]. However, in the recent study examining the effect of 0.04 mg kg$^{-1}$ per day CLR01 administered twice weekly in a different mouse model of PD, the brain-to-blood ratio of CLR01 was measured following subcutaneous administration and was found to be an order of magnitude higher than when the compound was administered intravenously[15], likely due to a slow release of the compound from the subcutaneous adipose tissue into the bloodstream.

The results presented here provide another missing piece of the puzzle. Assuming a similar behavior of CLR16 and CLR01, which is supported by the competition data presented in Fig. 9, rather than binding to every potential binding site in the cell, MTs are quickly confined into the very organelles into which the cells direct abnormal protein aggregates for clearance—endolysosomes and autolysosomes. As these organelles occupy roughly 1–2% of the cell volume, our data suggest that the effective concentration of MTs is could be two orders of magnitude higher in these compartments compared to what would be predicted if they diffused freely throughout the cell (and the extracellular space). Presumably, this allows MTs to disrupt the abnormal protein assemblies efficiently and facilitate their clearance even when the MTs are administered at a dose as low as 0.04 mg kg$^{-1}$, as was done in the AD[16] and PD[15] mouse models.

The colocalization experiments in tau-biosensor cells (Fig. 8) support this putative mechanism. The images show colocalization of CLR16 (underrepresented due to the low pH) with tau in lysosomes, but not with large tau aggregates outside the lysosomes, suggesting that within the lysosomal compartment, MTs can inhibit tau aggregation and dissociate small tau aggregates. As accumulation of aggregated tau and other amyloidogenic proteins has been shown to impair lysosome function[43,44], binding of MTs to these proteins in the lysosome apparently allows dissociation of their oligomers and aggregates and restoration of the lysosome function, thus accelerating the clearance of the aberrant assemblies. Indeed, a recent study in a mouse model of Sanfilippo syndrome type A showed massive lysosomal storage of amyloidogenic proteins including α-synuclein, Aβ, tau, and PrP in the brain, leading to mislocalization of the lysosomes in the neuronal perikarya and preventing their fusion with autophagosomes, which concentrated in the cell periphery and axons[26]. A daily

injection of CLR01 for 4.5 months restored lysosomal function and autophagy flux and ameliorated the neuroinflammation and memory deficits in this mouse model. Our findings that MTs colocalize with tau in lysosomes are consistent with these results and provide insight into the mechanism by which CLR01 restores lysosomal function.

Important questions remain to be answered, including the specific mechanisms that lead to accumulation of MTs in acidic compartments and the actual concentration of the molecules in these compartments. Nonetheless, our study delineates the ways by which MTs are internalized in living cells and become rapidly compartmentalized in organelles responsible for protein degradation, allowing the MTs to achieve high concentrations exactly where they are needed. This information supports the development of MTs as therapeutic drugs and may be used in future studies aimed at designing compounds with improved anti-amyloid activity.

## Materials and methods

**Synthesis of MTs.** CLR01 and CLR16 were prepared as described previously[25,45]. To synthesize CLR18, monophosphate monobutyl phosphate tweezer 1 (Supplementary Fig. 9a, 10 mg, 12.8 μmol) was dissolved in 2 mL 1:1 tetrahydrofuran–water (THF/$H_2O$) mixture in a 5 mL round-bottom flask together with the Gly-Lys-Azac-FAM peptide 2 (10.3 mg, 16 μmol). Freshly distilled diisopropylethylamine (11.3 μL) was added to a previously degassed solution. Subsequently, a copper sulfate solution (8.3 mg $CuSO_4 \cdot 5H_2O$, 33 μmol in 1 mL water) was mixed with a sodium ascorbate solution (13 mg, 66 μmol in 1 mL water) and the catalytic brew was immediately added to the reaction solution. The reaction mixture was stirred for 16 h at room temperature (RT). Quenching proceeded by addition of HCl (2.5 %, 5 mL, Supplementary Fig. 9a). Formation of a yellow precipitate was observed immediately after addition of the acid. The suspension was filtered directly through a D4-fritted funnel and the collected solid was subsequently washed with HCl (2.5%, 2 × 2 mL) and water (2 × 1 mL). The crude product was rinsed with distilled THF from the fritted funnel and dried on a rotary evaporator. Further purification proceeded via reverse-phase preparative HPLC (MeCN/$H_2O$ = 1/1 gradient). CLR18 was as obtained as a yellow solid (16 mg, 11.24 μmol, 88 % yield).

$^1$H NMR (600 MHz, DMSO-$d_6$, Supplementary Fig. 9b) δ 8.88 (s, 2H), 8.57 (s, 2H), 8.48 (s, 1H), 8.38 (s, 2H), 8.26 (d, $J = 8.1$ Hz, 1H), 8.06 (d, $J = 8.2$ Hz, 1H), 7.93 (s, 1H, triazol-H), 7.32 (s, 2H), 7.13–6.99 (m, 8H), 6.79–6.72 (m, 4H), 6.67 (s, 2H), 6.57 (s, 4H), 5.11 (s, 2H), 4.29 (s, 4H), 4.05 (t, $J = 13.0$ Hz, 5H), 3.95 (s, 2H), 3.73 (s, 2H), 3.57 (s, 1H), 2.83 (s, 3H), 2.37 – 2.10 (m, 10H), 2.00 (d, $J = 12.0$ Hz, 2H), 1.75 (s, 2H), 1.60 (s, 4H), 1.40 (s, 4H), 1.23 (s, 2H).

$^{13}$C NMR (151 MHz, DMSO-$d_6$) δ 168.70, 165.05, 160.23, 152.32, 150.75, 150.72, 150.67, 147.27, 136.86, 129.58, 126.96, 124.86, 124.62, 123.76, 121.88, 117.22, 116.99, 113.20, 102.73, 69.19, 68.14, 50.78, 50.75, 50.70, 48.48, 48.38, 41.15, 31.18, 25.60, 23.05.

HRMS; $m/z$ [M + H]$^+$: 1423.3752 calc., 1423.3790 obs.

**Fluorescence measurements of CLR16 and CLR18 at different concentrations and pH values.** Fluorescence spectra were recorded using a Hitachi F-4500 Fluorescence Spectrophotometer. The fluorescence emission was measured in the range 300–700 nm with excitation at 285 nm. Background spectra of buffer alone were subtracted from all the spectra, which therefore are presented as ΔI as a function of wavelength. Each data point is presented as mean ± SD of three technical replicates. For determining the fluorescence intensity change with concentration, CLR16 or CLR18 were dissolved in PBS (0.137 M NaCl, 0.0027 M KCl, 0.011 M sodium phosphate, pH 7.4) at a concentration of 12.5 μM and diluted serially in the same buffer.

For determining the dependence of the fluorescence intensity on the solution's pH, phosphate-citrate buffers at pH 3–8 were prepared by mixing 0.2 M $Na_2HPO_4$ and 0.1 M of citric acid and adjusting the pH using NaOH or HCl, as needed. CLR16 or CLR18 were dissolved in each buffer at a final concentration of 5 μM.

**NMR titration.** $^1$H NMR titration was performed using CLR16 (host) and Ac-Lys-OMe (guest) to monitor changes in the chemical shift of the Lys alkyl side chain caused by host/guest interactions. Six hundred microliters of 0.38 mM Ac-Lys-OMe in a 10 mM sodium phosphate, pH 7.4, was placed in an NMR tube and the $^1$H-spectrum was recorded. Keeping the guest concentration constant, CLR16 was titrated into the solution at concentrations ranging from 0.02 to 3.87 mM. The $^1$H NMR spectrum was recorded after each addition and the chemical shift changes of the Lys side chain methylene protons were determined.

**Host–guest fluorescence titration**. Fluorescence titrations were performed using the hosts CLR16 (3.32 μM) or CLR18 (7.76 μM) dissolved in 10 mM sodium phosphate, pH = 7.4, and the guest Ac-Lys-OMe to monitor fluorescence quenching of the tweezer moiety ($\lambda_{em}$ = 339–342 nm) upon host/guest interaction. The host concentration was kept constant. Seven hundred microliters of the host solution was placed in a quartz cuvette and the fluorescence spectrum was measured using $\lambda_{ex}$ = 284 nm. The guest solution was added stepwise up to 50–90 equivalents. After each addition, the fluorescence spectrum was recorded. ΔI at $\lambda_{max}$ (339–342 nm) was plotted against Ac-Lys-OMe concentrations and the binding constant was calculated using a nonlinear regression in SigmaPlot 10.

**Cell culture**. Undifferentiated human neuroblastoma SH-SY5Y cells were purchased from ATCC and were cultured in Dulbecco's modified Eagle's medium (DMEM)/F12 medium (Gibco, 11320033) supplemented with 10% fetal bovine serum (FBS) and a penicillin–streptomycin solution (100 units mL$^{-1}$ of penicillin and 100 μg mL$^{-1}$ of streptomycin, Caisson Labs, PSL01). Tau-biosensor cells[33] were obtained from ATCC and cultured in DMEM supplemented with GlutaMAX (4 mM of L-alanyl-L-glutamine, Gibco 10569044), 10% FBS, and penicillin–streptomycin. Cells were kept under a 5% $CO_2$ atmosphere at 37 °C. Cell lines were used as is and were not authenticated. The cell lines were not tested for mycoplasma contamination.

Experiments using primary cultures were carried out in accordance with National Research Council Guide for the Care and Use of Laboratory Animals, approved by the University of California-Los Angeles Institutional Animal Care Use Committee, and performed with strict adherence to the guidelines set out in the National Institutes of Health Guide for the Care and Use of Laboratory Animals.

Primary mouse hippocampal neurons were derived from postnatal day 1 wild-type (C57Bl/6 × C3H) F1 mice. Isolated hippocampi were incubated in a 0.5 mg mL$^{-1}$ papain/0.6 μg mL$^{-1}$ DNAase solution for 20 min at 37 °C. Then, the tissue was washed and triturated by pipetting 20 times through filtered 1000-μL tips. Dissociated cells were collected by centrifugation (200 × g, 3 min at 25 °C) and resuspended in complete medium comprising neurobasal medium (Gibco, 21103) supplemented with 2% B-27 Plus (Gibco A3582801), Antibiotic-Antimycotic (100 units mL$^{-1}$ of penicillin, 100 μg mL$^{-1}$ of streptomycin, and 250 ng mL$^{-1}$ of amphotericin B, Gibco 15240062), and GlutaMAX. Cells from both hippocampi of each mouse were pooled together and plated in two wells of an eight-well chambered coverglass (Thermo, 155409PK). The coverslips were coated using a 0.5 mg mL$^{-1}$ poly-ornithine solution (Sigma, P8638), followed by coating with 5 μg mL$^{-1}$ laminin (Corning, 354232) at RT for 16 h. Nonneuronal cells were removed by addition of 2 μM cytosine arabinoside (Acros Organics, AC449560010) after 72 h in culture. Antibiotics-depleted complete medium was replenished twice weekly, replacing 1/3 of volume each time. Cells were used for imaging on day 14 of the culture.

Mouse astrocytes were purified and cultured as described previously[46]. Three 150-mm diameter petri dishes were coated first with species-specific secondary antibodies[47] and then with antibodies against CD45 (BD 550539), a hybridoma supernate against the O4 antigen[47], and an antibody against HepaCAM (R&D Systems, MAB4108), respectively. Cerebral cortices were dissected from postnatal day 2 C57Bl/6 pups, and digested with papain to obtain a single-cell suspension. Microglia, macrophages, and oligodendrocyte precursor cells were depleted and astrocytes were purified by incubating the suspension sequentially on the CD45 antibody, anti-O4 hybridoma supernate, and HepaCAM antibody-coated petri dishes. Purified astrocytes were plated on poly-D-lysine coated plastic coverslips in a serum-free medium containing DMEM (Life Technologies, 11960069), Neurobasal (Life Technologies, 21103049), sodium pyruvate (Life Technologies, 11360070), SATO[48], glutamine (Life Technologies, 25030081), N-acetyl cysteine (Sigma, A8199), and heparin binding epidermal growth factor (Sigma, E4643).

**Fluorescence microscopy cell imaging**. Live-cell, time-lapse images were obtained using a BZ-X710 fluorescence microscope (Keyence). CLR16 or CLR18 were added immediately before initiating imaging and images were captured every 20 min. In experiments imaging CLR16 in the presence of CLR01, CLR01 was added 4 h before adding CLR16. Cell nuclei were stained with Hoechst dye. For lysosome and mitochondria labeling, LysoTracker™ Deep Red (Invitrogen, L12492) or MitoTracker™ Deep Red FM (Invitrogen, M22426) were added to the culture medium at final concentrations of 200 or 100 nM, respectively, 30 min before initiating imaging. To visualize the actin cytoskeleton, early endosomes, or late endosomes, actin-GFP, Rab5a-GFP, or Rab7a-GFP, respectively, were expressed in SH-SY5Y cells by treating the cells with the corresponding CellLight reagents (Invitrogen, C10582, C10586, C10588, respectively). These reagents are ready-to-use transient transfection mixtures of DNA constructs packaged in baculovirus. Six microliters of undiluted solution were added to 200 μL cell-culture medium per well. Cells were used for imaging after 16 h of incubation. To visualize autophagosomes, p62-GFP was expressed in SH-SY5Y cells using a Premo™ Autophagy Sensor p62-GFP Kit (Invitrogen, P36240). This reagent uses the same system as the CellLight reagents so the same concentration was used for GFP-p62 expression. Cells were pretreated with 60 μM of chloroquine for 5 h to let autophagosomes accumulate before adding CLR16. 3-D images and movies were created by

combining Z-stacked images using the 3-D image viewer of BZ-X analyzer software. In all cases, measurements from three fields of view in each experiment were used for statistical analysis.

**Endocytosis inhibition**. SH-SY5Y cells were transfected with the late-endosome marker GFP-Rab7a as described above. Sixteen hours after transfection, cells were treated with 0, 40, 80, or 120 μM of the endocytosis inhibitor dynasore (Abcam, ab120192) and 5 μM CLR16. Before imaging the cells, the culture medium was removed and the cells were washed with PBS to remove any CLR16 in the medium or on the surface of the cell. Cells were imaged every hour between 3 and 6 h. For quantitation, three areas were chosen randomly from different fields of view for each time point and each well. For collection of haze-reduced images, specific settings (Image/Image filter/Haze-reduction/setting-5) of the microscope were used to remove any background fluorescence including fluorescence in the cytoplasm, so that only bright, punctate fluorescence was visible. The percentage of CLR16 colocalized with Rab7a was calculated by dividing the number of yellow puncta (CLR16 in late endosomes/endolysosomes) by the total number of yellow and green puncta (Rab7a) × 100.

**Cytotoxicity**. Cell viability was measured using modifications of previously published protocols[12,49]. Briefly, SH-SY5Y cells were plated at $4 \times 10^4$ cells in 96-well plates in 100 μL of serum-supplemented medium. Dynasore at 20, 30, 40, 60, 80, 100, 110, or 120 μM was added to the cells in six technical replicates and incubated for 6 h. Control cells were incubated in the absence or dynasore. At the completion of the incubation period, 10 μL of PrestoBlue (Invitrogen) was added to each well and incubated for 10 min at 37 °C. Fluorescence was read with $\lambda_{ex}$ = 560 nm and $\lambda_{em}$ = 590 nm using a BioTek, Synergy HT plate reader.

**Tau seeding**. Tau seeds were prepared from PS19-mouse hippocampal extracts and added to biosensor cells for measurement of intracellular tau aggregation as described previously[19]. CLR16 was added 24 h after seeding.

**Fluorescence titration of CLR16 by CLR01**. CLR16 fluorescence were measured using an F-4500 Fluorescence Spectrophotometer (Hitachi). CLR16 was added to the cell-culture medium at a final concentration of 5 μM and increasing amounts of CLR01 were added and mixed by pipetting. The fluorescence emission was measured between 560 and 700 nm. Background emission from the cell-culture medium was subtracted. The peak emission was calculated as the average of the measurements between 573 and 575 nm.

**Image analysis**. Images were analyzed using BZ-X Analyzer (Keyence) and ImageJ[50]. For the quantification of fluorescence intensity, images were captured using appropriate filters and thresholds were set to select high fluorescence intensity areas. First, the organelle area was defined using the appropriate marker. When defining the cytoplasm area, the corresponding organelle areas were overlaid onto the actin-GFP images and excluded. The area selections then were overlaid onto the CLR16 image and the average CLR16 fluorescence intensity in those two sets of areas were measured. The measurement was done for each time point and the results were plotted against time. Nonlinear curves were fitted in Prism 7.0e, using the "One site – specific binding" algorithm. The plateau values were defined as the average of the measurements during the last 2 h. For quantifying the colocalization of MT puncta and organelles, first, the background of the images was subtracted using ImageJ's rolling ball algorithm. Next, JACoP plugin[51] for ImageJ was used to obtain the Manders' coefficients M1 and M2[28]. M1 shows the fraction of MT puncta overlapping organelles, whereas M2 shows the fraction of organelles overlapping MT puncta. Manders' coefficients range from 0 to 1, where 0 means no overlap and 1 means perfect overlap.

**Statistics and reproducibility**. Experiments were conducted at a minimum of three independent biological replicates unless otherwise indicated. The number of data points is indicated in the figure legends where appropriate. Statistical analysis was conducted using GraphPad Prism 9.2 unless otherwise indicated.

**Reporting summary**. Further information on research design is available in the Nature Research Reporting Summary linked to this article.

## Data availability
Source data for all the graphs included in this paper are available as Supplementary Data in Excel format. All other data are available from the corresponding author upon reasonable request.

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

## Acknowledgements

The work was supported by NIH/NIA grants R01 AG050721 and RF1 AG054000 and the Deutsche Forschungsgemeinschaft DFG provided generous funding of the Collaborative Research Centre CRC 1093 Supramolecular Chemistry on Proteins.

## Author contributions

G.B. and Z.L. designed the study. I.H. and A.K. synthesized the fluorescently labeled MTs. J.L. and Y.Z. provided mouse primary astrocytes. Z.L., I.S., and A.K. performed the experiments. Z.L., I.S., A.K., T.S., and G.B. analyzed the data and wrote the manuscript. F.-G.K. and T.S. provided materials and critical reading of the manuscript.

## Competing interests

G.B. and T.S. are inventors of International Patent PCT/EP2010/000437, US Patent No. US 8481484 B2, European Patent No. EP 2493859 A1, and International Patent Application No. PCT/US2019/039943, US Patent Application No. 17/255,963 protecting the composition of matter and methods of use of MTs. The remaining authors declare no competing interests.
