## [Peer Review File · Communications Biology]

Reviewers' comments:

Reviewer #1 (Remarks to the Author):

This manuscript deals with the molecular tweezers CLR16 and CLR18, fluorescent analogs of CLR01, a molecular tweezers binding selectively to lysine residues and exhibiting unique pharmacological properties. The intrinsic fluorescent properties of CLR16 and CLR18 enable intracellular trafficking studies and mechanism elucidation.

In summary, CLR16 is internalized mainly via a dynamin-dependent pathway and ends in late endosomes and lysosomes, as well as in autophagosomes, in a lower extent. Some diffuse fluorescence suggests also a cytoplasmic delivery of CLR16. These findings were confirmed in several cell lines (SH-SY5Y, primary mouse hippocampal neurons, primary astrocytes) and with CLR18.

In addition, CLR16 was found to colocalize with tau oligomers in the lysosomes, suggesting that their low therapeutic concentration could be counterbalanced with lysosome accumulation, where they might destabilize tau small aggregates and restore lysosomal function.

This study is well conducted, well-explained, and significantly improve the knowledge of CLR01 mechanism. CLR01 is a promising molecule, it is the first molecular tweezer exhibiting such a broad range of therapeutic efficacy (inhibition of protein aggregation and antiviral efficacy) without causing significant toxicity, which is quite unexpected. Therefore, deepening the knowledge on this molecule is crucial to push forward its clinical translation. This study brings significant progress to this project, and is also pioneering the field of molecular tweezers pharmaceutical applications.

My concerns are the following:

1. CLR16 and CLR18 are CLR01 appended with a fluorescent probe. As seen the structure, the molecular weight of molecular tweezers, its charge (TAMRA is positively charged, the 6-FAM has a pendant carboxylic acid group) and their hydrophobicity should be strongly impacted. This could also impact their binding properties with the proteins, and with the cell membrane and trafficking. Although the authors are aware of this issue and address it by confirmation studies of CLR18 (Fig 4) and competition with CLR01 (Fig 10), I would suggest a binding study with CLR16 and CLR18 as compared to CLR01, for instance with Tau protein. It is mentioned that "As neither target binds avidly to any target" (line 411). Has the binding of fluorescent compounds been checked previously? I saw the synthesis reported in Mol Ther 28, 1167-1176, but I could not see any binding experiment. Alternatively, a protein aggregation inhibition experiment could also be conducted, comparing CLR01, CLR16 and CLR18. This would also confirm the colocalization studies of tau small aggregates (Fig 9).
2. Why the structure of CLR18 bear a pendant carboxylic group? It is not required for 6-FAM fluorescence, or in the MT structure?
3. For qualitative microscopy studies, is it possible to provide additional images in the supporting information, as replicates?
4. For colocalization studies, would it be possible to quantify colocalization? for instance in figures 3, 4, 5, 8, 9.

Reviewer #2 (Remarks to the Author):

In this manuscript, Gal Bitan and coworkers have designed and synthesized new fluorescently labeled molecular tweezers (MTs) to explore their internalization, intracellular localization, and mechanism of endocytosis in brain cells including neurons and astrocytes. The authors showed that these compounds are internalized in neurons and astrocytes, at least partially through dynamin-dependent endocytosis. In addition, the authors demonstrated that the molecular tweezers concentrate rapidly in acidic compartments, primarily lysosomes. Accumulation of molecular tweezers in lysosomes may occur both through the endosomal-lysosomal pathway and via the autophagy-lysosome pathway. Moreover, by visualizing colocalization of molecular tweezers, lysosomes, and tau aggregates, the authors showed that lysosomes likely are the main

site for the intracellular anti-amyloid activity of molecular tweezers.

It should be noted that, this study delineates the ways by which MTs are internalized in living cells and become rapidly compartmentalized in organelles responsible for protein degradation, allowing the MTs to achieve high concentrations exactly where they are needed. This information supports the development of MTs as therapeutic drugs and may be used in future studies aimed at designing compounds with improved anti-amyloid activity. Hence, I believe this manuscript will be well prepared and organized and interesting to the supramolecular chemistry and biological science. In my opinion, it could be accepted for the publication on Communications Biology after some points below are addressed:

1. As a control experiment, concentration-dependent fluorescence intensity of the CLR16 in vitro (the pH value can be maintained at 7.4 in the buffer solution) experiment should be supplemented.
2. In addition, the pH-dependent fluorescence intensity of CLR16 in vitro at specific concentration experiments should be supplemented.
3. Is it possible to estimate the concentration of the MTs at the specific area in the cells by measuring the fluorescent intensity?

Reviewer #3 (Remarks to the Author):

In this new manuscript, authors continue investigation into the mechanism of action of Lys-selective molecular tweezers, a novel class of protein aggregation antagonist with established binding selectivity in vitro and efficacy in biological models. A major mechanistic question has been how tweezers generate high potency in biological models despite having weak binding affinity for aggregation prone protein targets. This manuscript tests the hypothesis that the tweezers are potent because they accumulate within specific subcellular compartments. It does so by tracking the uptake and intracellular localization of two highly fluorescent tweezer analogs (generated by click chemistry). Authors claim the results add a missing piece of the puzzle by demonstrating efficient uptake into cells and preferential concentration in acidic compartments such as lysosomes, and postulate that tweezer activity is exerted on targets in this compartment. This possibility is exciting because the endo/lysosomal system has been implicated in tau seed uptake. Overall, the results add important information to this class of aggregation antagonist, and the work is appropriate for the journal. Specific comments are summarized below.

Specific comments

1. Fig. 4: text states that "overlap between CLR16 and the dye LysoTracker™ was nearly complete", and that "CLR18 showed that nearly every punctum of the compound in SH-SY5Y cells overlapped with a lysosome". But panel 4DEF shows highly uneven colocalization
2. Fig. 6: cellular uptake assay leverages dyansore, a compound that can influence results by affecting cell viability. Has viability been assessed as part of Fig. 6 experimentation?
3. Fig. 9: colocalization of tweezers with seeded tau aggregates is shown in biosensor cells. Authors should clarify the conditions of this experiment, especially whether the fluorescence represents FRET between YFP and CFP 4RD tau or just YFP fluorescence as labeled on the figure. The latter may not represent seeding and would require additional controls to confirm seeding.
4. Results section "CLR16 colocalizes with tau aggregates in lysosomes". Last sentence speculating on "MTs facilitating tau clearance by dissociating tau oligomers" does not belong in Results section as no experimentation address oligomers or their dissociation.

Reviewer #1 (Remarks to the Author):

This manuscript deals with the molecular tweezers CLR16 and CLR18, fluorescent analogs of CLR01, a molecular tweezers binding selectively to lysine residues and exhibiting unique pharmacological properties. The intrinsic fluorescent properties of CLR16 and CLR18 enable intracellular trafficking studies and mechanism elucidation.

In summary, CLR16 is internalized mainly via a dynamin-dependent pathway and ends in late endosomes and lysosomes, as well as in autophagosomes, in a lower extent. Some diffuse fluorescence suggests also a cytoplasmic delivery of CLR16. These findings were confirmed in several cell lines (SH-SY5Y, primary mouse hippocampal neurons, primary astrocytes) and with CLR18.

In addition, CLR16 was found to colocalize with tau oligomers in the lysosomes, suggesting that their low therapeutic concentration could be counterbalanced with lysosome accumulation, where they might destabilize tau small aggregates and restore lysosomal function.

This study is well conducted, well-explained, and significantly improve the knowledge of CLR01 mechanism. CLR01 is a promising molecule, it is the first molecular tweezer exhibiting such a broad range of therapeutic efficacy (inhibition of protein aggregation and antiviral efficacy) without causing significant toxicity, which is quite unexpected. Therefore, deepening the knowledge on this molecule is crucial to push forward its clinical translation. This study brings significant progress to this project, and is also pioneering the field of molecular tweezers pharmaceutical applications.

We thank the reviewer for their positive evaluation of the quality and significance of the study.

My concerns are the following:

1. CLR16 and CLR18 are CLR01 appended with a fluorescent probe. As seen the structure, the molecular weight of molecular tweezers, its charge (TAMRA is positively charged, the 6-FAM has a pendant carboxylic acid group) and their hydrophobicity should be strongly impacted. This could also impact their binding properties with the proteins, and with the cell membrane and trafficking.

Although the authors are aware of this issue and address it by confirmation studies of CLR18 (Fig 4) and competition with CLR01 (Fig 10), I would suggest a binding study with CLR16 and CLR18 as compared to CLR01, for instance with Tau protein. It is mentioned that “As neither target binds avidly to any target” (line 411). Has the binding of fluorescent compounds been checked previously? I saw the synthesis reported in Mol Ther28, 1167-1176, but I could not see any binding experiment. Alternatively, a protein aggregation inhibition experiment could also be conducted, comparing CLR01, CLR16 and CLR18. This would also confirm the colocalization studies of tau small aggregates (Fig 9).

In very recent experiments, tweezer-peptide conjugates with and without a FAM label were added to 14-3-3 proteins and produced very similar nanomolar in FP titrations. Due to the presence of the peptide moiety, these constructs were highly selective for the central protein cleft as evidenced by X-ray crystallography and produced clean 1:1 complexes. Direct titrations between FAM derivatives and 14-3-3 proteins indicated negligible affinity in both FP as well as ITC titrations. CLR16 and CLR18 do not contain a peptide binding unit and therefore bind to

any protein with accessible Lys residues with high tweezer:protein stoichiometry, rendering quantitative binding experiments very difficult. There are 37-44 Lys residues in tau, depending on the isoforms, so the chances of getting a clear binding curve are very small.

Aggregation experiments using tau are problematic because the protein does not aggregate on its own. Aggregation may be induced by polyanions, such as heparin, but unfavorable interaction with CLR01 leads to a bimodal inhibition behavior (Despres et al, ACS Chem. Biol. 2019), making interpretation of such experiments difficult. In addition, the TAMRA and FAM groups will most likely affect ThT fluorescence, which is used to measure the aggregation kinetics.

Therefore, we decided to carry out direct titration experiments of both fluorescent tweezers with N-acetylated lysine esters as small peptide models to demonstrate that they operate just the same way as the parent compound and include a peptidic lysine sidechain inside their open cavity.

Our first attempt was to use NMR titrations between CLR16 and Ac-Lys-OMe to demonstrate binding, as upfield shifts of the Lys sidechain methylene groups are highly indicative of inclusion of the sidechain in the tweezer's cavity. Unfortunately, the NMR spectra showed substantial line broadening during the titration. However, a very interesting observation was made: Between 0 and 1 equivalents, the Lys protons did not shift but intensities of the free side chain decreased and finally vanished at the equivalence point. At the same time, a new set of upfield-shifted signals appeared gradually, indicating insertion into the tweezer cavity with slow exchange on the NMR time scale. Since NMR titrations require high concentrations in the mM range, this behavior, together with the observed line broadening suggests formation of oligomers or aggregates.

In view of this behavior, and because the cell-culture experiments reported in the paper were done using low μM concentrations of the fluorescent tweezers, we conducted fluorescence titrations of CLR16 and CLR18 with Ac-Lys-OMe at low μM concentrations. This time distinct binding curves were obtained with significant quenching of the tweezer's fluorescence at 330 nm (distinct from the fluorescence of the TAMRA or FAM groups), strongly indicative of Lys inclusion. Interestingly, the binding curves displayed a biphasic behavior: An initial very strong binding step followed by a second weak binding of an additional Ac-Lys-OMe molecule, likely to the fluorescent tag. The 2:1 stoichiometry of each tweezer binding to two Ac-Lys-OMe indicates the existence of single tweezer molecules and rules out aggregation at micromolar concentrations. The NMR titration with CLR16 and two very similar fluorescence titrations of CLR16 and CLR18 are shown as Supplementary Figures 2–4 and are discussed in the main text of the revised manuscript.

2. Why the structure of CLR18 bear a pendant carboxylic group ? It is not required for 6-FAM fluorescence, or in the MT structure ?

The pendant carboxylic group in the structure of CLR18 is neither required for the fluorescence label nor for the MT function. It is just a consequence of our strategy to obtain a clickable FAM label. Contrary to the TAMRA fluorophore, azido-labeled FAM was not commercially available at the time the compound was prepared for this project. Therefore, we used FAM-lysine, incorporated this into a small peptide and attached a terminal azide. Specifically, we started with a glycine-loaded Wang resin, coupled FAM-labeled lysine, and then azido-acetic acid. Cleavage from the resin released the free carboxylic acid at the C-terminus of this construct, which

actually increases the water solubility of the tweezer conjugate. An inherent advantage of this strategy is the introduction of a peptidic spacer between tweezer and fluorophore. The SPPS strategy is outlined below.

For CLR16 we purchased a TAMRA derivative with a terminal azide, allowing straightforward click reaction with an alkyne tweezer.

3. For qualitative microscopy studies, is it possible to provide additional images in the supporting information, as replicates ?

Additional images have been added as supplementary information in all cases and the sentence “The supplementary images here and in subsequent figures have been included simply to demonstrate the reproducibility of the data” was added to the second paragraph of the Results section. In our opinion, in view of the quantitation added as Supplementary Figure 5 in response to the reviewer’s comment 4, which demonstrates the degree of reproducibility of the data, these supplementary figures are superfluous, and our preference is to leave them out of the final version of the manuscript. We would appreciate getting the Editor’s input on this point.

4. For colocalization studies, would it be possible to quantify colocalization ? for instance in figures 3, 4, 5, 8, 9.

We thank the reviewer for this suggestion. The colocalization was quantified and the data are shown in a new Supplementary Figure 5.

Reviewer #2 (Remarks to the Author):

In this manuscript, Gal Bitan and coworkers have designed and synthesized new fluorescently labeled molecular tweezers (MTs) to explore their internalization, intracellular localization, and mechanism of endocytosis in brain cells including neurons and astrocytes. The authors showed that these compounds are internalized in neurons and astrocytes, at least partially through dynamin-dependent endocytosis. In addition, the authors demonstrated that the molecular tweezers concentrate rapidly in acidic compartments, primarily lysosomes. Accumulation of molecular tweezers in lysosomes may occur both through the endosomal-lysosomal pathway and via the autophagy-lysosome pathway. Moreover, by visualizing colocalization of molecular tweezers, lysosomes, and tau aggregates, the authors showed that lysosomes likely are the main site for the intracellular anti-amyloid activity of molecular tweezers.

It should be noted that, this study delineates the ways by which MTs are internalized in living cells and become rapidly compartmentalized in organelles responsible for protein degradation, allowing the MTs to achieve high concentrations exactly where they are needed. This information supports the development of MTs as therapeutic drugs and may be used in future studies aimed at designing compounds with improved anti-amyloid activity. Hence, I believe this manuscript will be well prepared and organized and interesting to the supramolecular chemistry

and biological science. In my opinion, it could be accepted for the publication on Communications Biology after some points below are addressed:

We thank the reviewer for the positive evaluation of the manuscript.

1. As a control experiment, concentration-dependent fluorescence intensity of the CLR16 in vitro (the pH value can be maintained at 7.4 in the buffer solution) experiment should be supplemented.

We thank the reviewer for this suggestion. We have added the suggested concentration-dependence of both CLR16 and CLR18 in the revised manuscript (Supplementary Figure 1A, B) and a section in the beginning of the Results describing these results.

2. In addition, the pH-dependent fluorescence intensity of CLR16 in vitro at specific concentration experiments should be supplemented.

We thank the reviewer in particular for this suggestion, which has helped gaining new insight into the behavior of the compounds. The data also have been added to the revised manuscript (Supplementary Figure 1C, D) and to the first section of the Results. They reveal that the fluorescence of CLR16 decreases when the pH decreases as the compound traffics from the cytoplasm to the endosomes and lysosomes, indicating that the fluorescence we measured is an underrepresentation of the actual concentration of CLR16. In the case of CLR18, a strong red shift from 520 to 570 nm occurs below pH 5, providing an explanation for the peculiar behavior of CLR18 pointed out by Reviewer 3 (point 1).

3. Is it possible to estimate the concentration of the MTs at the specific area in the cells by measuring the fluorescent intensity?

We agree with the reviewer that this is an important question. However, answering this question is technically difficult. To convert the fluorescence values to concentration requires creating a calibration curve, as the reviewer suggested in their comment 1 above. However, as this calibration curve is created in buffer and the exact impact of the cellular and organellar environments on the fluorescence is not known. As the new pH-dependence data demonstrate, the impact of the environment can be quite strong and its precise measurement in various cellular environments is beyond the scope of the manuscript. Therefore, we believe it is best to use a conservative approach and limit the data presentation and interpretation to the comparative data in Figures 6, 7, and 10, rather than attempting to estimate the absolute concentration of the compounds in the different organelles.

Reviewer #3 (Remarks to the Author):

In this new manuscript, authors continue investigation into the mechanism of action of Lys-selective molecular tweezers, a novel class of protein aggregation antagonist with established binding selectivity in vitro and efficacy in biological models. A major mechanistic question has been how tweezers generate high potency in biological models despite having weak binding affinity for aggregation prone protein targets. This manuscript tests the hypothesis that the

tweezers are potent because they accumulate within specific subcellular compartments. It does so by tracking the uptake and intracellular localization of two highly fluorescent tweezer analogs (generated by click chemistry). Authors claim the results add a missing piece of the puzzle by demonstrating efficient uptake into cells and preferential concentration in acidic compartments such as lysosomes, and postulate that tweezer activity is exerted on targets in this compartment. This possibility is exciting because the endo/lysosomal system has been implicated in tau seed uptake. Overall, the results add important information to this class of aggregation antagonist, and the work is appropriate for the journal.

We thank the reviewer for their positive evaluation of the importance of the work.

Specific comments are summarized below.

Specific comments

1. Fig. 4: text states that “overlap between CLR16 and the dye LysoTracker™ was nearly complete”, and that “CLR18 showed that nearly every punctum of the compound in SH-SY5Y cells overlapped with a lysosome”. But panel 4DEF shows highly uneven colocalization

We thank the reviewer for pointing out this difference between the behavior of CLR16 and CLR18. The new pH-dependence experiments suggested by Reviewer 2 have helped understanding the peculiar behavior of CLR18, revealing that at a pH between 4 and 5, which is the pH range of lysosomes, a strong red shift occurs, leading to the compound becoming undetectable using the green filter of the fluorescence microscope. We have added these data in the first section of the Results and in Supplementary Figure 1D. In addition, we added a new explanation of the data in the section describing the data in Figure 4.

2. Fig. 6: cellular uptake assay leverages dyansore, a compound that can influence results by affecting cell viability. Has viability been assessed as part of Fig. 6 experimentation?

We thank the reviewer for bringing up this important point. We have added Supplementary Figure 8, which demonstrates that under the experimental conditions used, Dynasore does not reduce the viability of the SH-SY5Y cells.

3. Fig. 9: colocalization of tweezers with seeded tau aggregates is shown in biosensor cells. Authors should clarify the conditions of this experiment, especially whether the fluorescence represents FRET between YFP and CFP 4RD tau or just YFP fluorescence as labeled on the figure. The latter may not represent seeding and would require additional controls to confirm seeding.

The fluorescence measurement in this experiment was of YFP only. The goal of the experiment was not to measure seeding but rather to test whether CLR16 colocalizes with tau puncta. Seeding of similar brain extracts from P301S-tau (PS19) mice recently was demonstrated in Di et al., *Alzheimer's Research & Therapy*, 2021.

4. Results section “CLR16 colocalizes with tau aggregates in lysosomes”. Last sentence speculating on “MTs facilitating tau clearance by dissociating tau oligomers” does not belong in Results section as no experimentation address oligomers or their dissociation.

We respectfully submit that although this sentence is in the Results section, it clearly is a speculative interpretation of the data rather than presentation of facts and therefore it is our preference not to change this sentence.

REVIEWERS' COMMENTS:

Reviewer #1 (Remarks to the Author):

The revised manuscript has gained in quality and quantity of results. I appreciate the integrity of authors describing their attempt and difficulties for comparing CLR01, CLR16 and CLR18. NMR and fluorescence studies have been added to support the affinity of CLR16 and CLR18 for lysine residue.

Additional images and quantification of colocalization were provided.

PH titrations gave further insight in the fluorescence behaviour of CLR18 in cells.

In my opinion, the current version is suitable for publication.

Reviewer #2 (Remarks to the Author):

The authors have addressed all of the issues. I suggest the acceptance of the manuscript in its current form.

Reviewer #3 (Remarks to the Author):

This reviewer's concerns have been addressed.